# An Approximation for Metal-Oxide Sensor Calibration for Air Quality Monitoring Using Multivariable Statistical Analysis

**DOI:** 10.3390/s21144781

**Published:** 2021-07-13

**Authors:** Diego Sales-Lérida, Alfonso J. Bello, Alberto Sánchez-Alzola, Pedro Manuel Martínez-Jiménez

**Affiliations:** 1Department of Automation Engineering, Electronics and Computer Architecture and Networks, University of Cádiz, 11519 Cádiz, Spain; pedromanuel.martinez@uca.es; 2Department of Statistic and Operations Research, University of Cádiz, 11510 Cádiz, Spain; alfonsojose.bello@uca.es (A.J.B.); alberto.sanchez@uca.es (A.S.-A.)

**Keywords:** air quality, metal-oxide sensor, monitoring, multivariable regression models, model calibration

## Abstract

Good air quality is essential for both human beings and the environment in general. The three most harmful air pollutants are nitrogen dioxide (NO_2_), ozone (O_3_) and particulate matter. Due to the high cost of monitoring stations, few examples of this type of infrastructure exist, and the use of low-cost sensors could help in air quality monitoring. The cost of metal-oxide sensors (MOS) is usually below EUR 10 and they maintain small dimensions, but their use in air quality monitoring is only valid through an exhaustive calibration process and subsequent precision analysis. We present an on-field calibration technique, based on the least squares method, to fit regression models for low-cost MOS sensors, one that has two main advantages: it can be easily applied by non-expert operators, and it can be used even with only a small amount of calibration data. In addition, the proposed method is adaptive, and the calibration can be refined as more data becomes available. We apply and evaluate the technique with a real dataset from a particular area in the south of Spain (Granada city). The evaluation results show that, despite the simplicity of the technique and the low quantity of data, the accuracy obtained with the low-cost MOS sensors is high enough to be used for air quality monitoring.

## 1. Introduction

Good air quality is essential for both humanity and the natural environment. Economic activities such as energy production, industry and agriculture, as well as the dramatic rise in traffic, release air pollutants into the environment that can lead to serious problems for our health [1]. In fact, the poor quality of air is the cause of more than 400,000 premature deaths in Europe each year, as well as a decrease in quality of life by causing or exacerbating asthma and respiratory problems [2,3].

There are several pollutants involved in air quality characterization, such as SOx, CO, NOx, O_3_, or particulate matter pollution [4,5]. From all of them, three of the most harmful air pollutants, in terms of damage to ecosystems, are nitrogen dioxide (NO_2_), ozone (O_3_) and particulate matter (specifically PM2.5, which is directly related to traffic) [6,7,8,9]. Thus, it is very important to monitor and analyze these elements in the air, especially in towns and cities, in order to detect dangerously high levels and take actions to reduce pollution [10,11].

In this regard, several agencies around the world are responsible for the air quality monitoring of their corresponding regions, such as the European Environment Agency (EEA) in Europe, or the Environmental Protection Agency (EPA) in the United States. Their data give relevant and reliable information to policymaking agents [12]. In particular, the use of air quality models to assess the potential changes in urban air quality concentrations is a fundamental element of air quality management. In this type of modeling, the input data require high spatio-temporal resolution to capture the variability in the urban environment. However, one of the main technical difficulties nowadays is the lack, or low quality, of input data on concentrations [13]. Due to the high cost of monitoring stations, only a few examples of this type of infrastructure have been deployed in cities, providing limited spatial coverage [14].

In order to address this problem, recent environmental agencies’ reports suggest that cities should participate in the input data acquisition, complementing official monitoring data with additional measurements of local air quality [13]. In this sense, the cities are increasingly aware of the potential for low-cost ‘citizen science’ sensors to help support the results of their air quality modeling [15,16]. These sensors offer air pollution monitoring at a lower cost and smaller size than conventional methods, making it possible for them to be installed in many more locations [17,18,19]. However, the accuracy of input data in air quality modeling is as important as the quantity of measures. Thus, the use of citizen science and citizen participation in air quality monitoring by means of these low-cost sensors is only feasible if they can provide accurate information [20,21].

Currently, the three most popular types of low-cost air quality sensors are electrochemical sensors (EC), metal-oxide sensors (MOS) and photoionization detectors (PID) [22,23]. Since the objective is to achieve the widest possible distribution of air monitoring sensors in cities, their price is an essential factor. In this sense, the cost of EC and PID sensors is prohibitive for most consumers (they can cost more than EUR 100). On the contrary, the cost of MOS sensors, which are usually below EUR 10, as well as their small dimensions, make them an excellent option for use by citizens [24,25]. However, it should be noticed that, in air quality monitoring, the pollutant concentration that sensors should capture is usually very small: in the order of parts per billion (ppb) or “μg/m^3^”. In this sense, the World Health Organization (WHO) casts some doubts on the reliability of low-cost sensors when the calibration methods provided by manufacturers are employed, because these methods may be questionable regarding very low concentrations [26]. Thus, the WHO, as well as the EEA [10], only recommends the use of these devices for air quality monitoring through an exhaustive calibration process and subsequent precision analysis [27,28].

In most of the works in the literature, sensor calibration is performed under laboratory conditions [29,30,31]. In this type of approach, controlled environments are created by injecting known concentrations of the specific pollutants to be measured. However, in these ideal laboratory conditions, other variables that are present in real environments are not taken into account. On the one hand, there could be particles of other components that are different from the pollutants to be measured in the specific region where the sensors should be used which are not considered in a laboratory. On the other hand, although other environmental factors in the specific region can be simulated in a laboratory, such as the temperature and relative humidity of the air, they may differ from the actual conditions [32].

In order to face these problems, several on-field calibration techniques have been proposed in the literature [33,34,35,36], which are based on the data obtained from the monitoring stations of the regional government agencies. This way, sensors are calibrated using the specific environmental conditions of the region where they will be used, and are, therefore, adapted to its temperature, humidity and air composition. In most of these works, the proposed calibration techniques are complex and not very intuitive, and they are applied by experts in the field. In addition, in those studies, a large amount of calibration data is available from sensors, since they have been placed close to the reference monitoring stations for long periods of time. However, we should remember that the objective of these low-cost sensors is the use of citizen science and citizen participation in air quality monitoring. Thus, in real situations, the sensors will be calibrated by field workers who are usually not so expert in applying complex techniques, and the available data for calibration may be limited, since locations close to monitoring stations cannot be used for long periods of time.

In this work, we present an on-field calibration technique for low-cost MOS sensors that tries to solve both problems commented on above: it can be easily applied by non-expert operators, and it can be used even with only a small amount of calibration data. The proposed technique is based on the well-known regression analysis tool [37,38,39], which is widely used for data modeling in a great variety of fields. In our approach, we have studied the different kinds of regression techniques in the literature, and we have selected the most appropriate one, taking into account the number of independent variables, the type of dependent variables and the shape of the regression curve. We apply and evaluate this technique with a real dataset from a particular area in the south of Spain (Granada city). The training and test data were used to fit and validate the model, respectively, using the R software [40]. The evaluation results show that, despite the simplicity of the technique and the low quantity of data, the accuracy obtained with the low-cost MOS sensors is high enough to be used for air quality monitoring. In addition, the proposed method is adaptive, in the sense that the calibration can be refined as more data become available.

The rest of the paper is organized as follows. In Section 2, we briefly present the sensors that are usually employed to measure the air pollutant concentrations, giving more details to the low-cost MOS used in this work, we describe and analyze the dataset used to validate the calibration technique, and we explain the calibration methodology. In Section 3, we apply this methodology to fit the pollutant concentrations corresponding to ozone (O_3_), nitrogen dioxide (NO_2_) and carbon monoxide (CO). The obtained results are statistically studied and discussed in Section 4, while Section 5 contains the main conclusions of this paper.

## 2. Material and Methods

### 2.1. Sensors

Before going into details about the sensors used in this work to measure air pollutant concentrations, we should clarify that the unit selected to express these concentrations will be “μg/m^3^” because this is the form used by the European Commission for regulation in the European framework. 

The European air quality standards set by the Ambient Air Quality Directive (EU, 2008) for the protection of human health [41], the air quality guidelines (AQGs) set by the World Health Organization (WHO) [42], and their subsequent revisions, define several aspects of values for the different pollutants, like typical qualitative levels, the averaging period, the time by which limit values can be overcome in a year, or alert values. In Spain, there are certain laws that refer to these standards; the most recent of their revisions were passed on 28 January 2011 in the form of the directive RD102/2011. Table 1 shows some of its aspects.

As mentioned in Section 1, in this study we have proposed the use of MOS sensors, since they are the most accessible to users from an economic point of view. These sensors are composed of a semiconductor layer, generally, tin dioxide (SnO_2_), which makes them especially sensitive to other oxides, and, by controlling the doping of the semiconductor, it is possible to make the material more sensitive to certain parameters. Therefore, when there are higher concentrations of these parameters in sampled air, the conductivity of this layer changes its values. It is worth mentioning that this conductivity keeps a direct relation with temperature, and, in general terms, they change in a proportional form. In addition, it should be noticed that, after a certain temperature, the sensibility to target gases can decrease, negatively affecting the quality of sensor detection. To take advantage of this property, electrodes are inserted into the detection layer of the sensor in order to increase its temperature in a controlled way (by using a heating circuit, such as a voltage divider with resistors) [43,44,45]. 

In particular, the MOS sensors used in this work are the ones incorporated in the devices developed in the “EcoBici (Kers bike)” research project (file number G-GI3002/IDIC) which resulted in a patented invention, application number P201600319 and publication number ES2638715 [46]. These devices were designed to take air quality values, accumulating the data and being able to configure the time in which the averages are sent to a web server, in real time, through the deployment of a sensor network using XBee technology (protocol ZigBee). The parameters measured by these devices are CO, O_3_ and NO_2_. It should be noticed that these sensors are non-specific sensors since they can measure other gases apart from the main gas [43], but these secondary gases are not those considered in this paper. It is worth mentioning that O_3_ and NO_2_ are linked by the Leighton relationship. Nevertheless, the proposed methodology is not affected by this relationship since it is already considered in the parameter estimation.

For the calibration tests, the devices were adapted to send the temporal average of the three parameters every 10 min in order to be synchronized with the calibration equipment. Figure 1 shows the three sensors incorporated in EcoBici end devices, which include an MQ-7 sensor for CO measuring [43], an MQ-131 sensor for O_3_ [44] and an MiCS-2714 sensor for NO_2_ [45].

The concentration values given by the curves in datasheets [43,44,45] are much higher than the values that should be measured in terms of air quality. Although some of the sensor manufacturers guarantee that the device is able to detect the presence of gas at tens of ppb, our own experience can confirm the information from the WMO, cited in Section 1, and discourage the use of these curves for low concentrations.

In order to carry out the measurement campaign for field calibration, we used the highly sophisticated equipment located in the sampling stations belonging to the Environment Council of the Andalusian government. In these sampling stations, which are mostly composed of measurement analyzers, the pollutant concentrations are taken continuously, 24 h/day, 365 days per year, except for breakdowns. The cost of this type of equipment generally exceeds the barrier of EUR 10,000, and it is used to analyze a single parameter. It should be noted that each autonomous community or region has its own criteria to collect the data. In the case of Andalusia, the analyzers used in their stations take a sample of the ambient air, previously conditioned and homogenized, and analyze it in periods ranging from 10 s to 10 min, depending on the pollutant to be analyzed. This information is averaged in 10 min periods, stored and published by the Spanish Ministry of Air Quality [47], and on the Andalusian Council website (available from the following day) [48].

In order to select the most suitable sampling stations for calibration campaigns, several factors should be taken into account, such as the latest calibration reviews of the station, accessibility, and measurements range obtained of the different parameters in the station in several days. Regarding the data range, it is highly important to choose a station that can provide a wide range of values in the different parameters to be calibrated. For example, if a station where quantitative O_3_ values do not exceed 50 ppb after several days is selected, the sensor may not be properly calibrated for higher concentration values. According to this criterion, a station localized in Granada city was selected from more than 100 Andalusian Council monitoring stations. Figure 2 shows a photo of the Granada sampling station, where it is possible to identify the EcoBici devices on it, next to the station analyzers.

Finally, it is important to take into account the particular conditions of temperature and humidity in Granada city, since both parameters affect the best adjustment of sensors, as will be seen in the data section. In fact, both parameters were requested by the agency in charge of the sampling station after the measurement campaign. In any case, if these data could not be obtained from the corresponding agency, another option would be to place temperature and humidity sensors in the devices.

### 2.2. Description of Dataset

The real dataset of the work in the present paper involves measurements, taken by both analyzers and sensors, of three particular gaseous pollutants: ozone, nitrogen dioxide and carbon monoxide, in addition to temperature and humidity measurements by the agency. The observations are collected in 490 registers which were taken from midnight, 00:00 h, 08/05/2016 until 09:30 h, 11/05/2016, at a ten-minute frequency. The respective pollutant variables corresponding to the analyzers, from now on also called patterns, have been denoted as “O_3_”, “NO_2_” and “CO”, the respective pollutant variables corresponding to the sensors as “O_3_s”, “NO_2_s” and “COs”, the temperature variable as “temp”, and the humidity variable as “hum”. To obtain a better fit of the models, we have added a new variable, called “COsR”, which is a version of COs without trend. The rectified COsR time series has been obtained by the ratio of the sensor values and its adjusted least squares regression line. Moreover, we have translated the time series to the sensor range modifying the scale. Therefore, finally, we count 9 variables of work in the dataset: temp, hum, O_3_, NO_2_, CO, O_3_s, NO_2_s, COs and COsR. 

The following sections show how to predict the pattern values for the gaseous pollutants O_3_, NO_2_ and CO, applying multivariable regression models and selecting the best fit by using the measurements of the sensors, O_3_s, NO_2_s and COs, and the values of temperature and humidity. That is, a general expression of the model would be:Y = f(X_1_, X_2_, …, X_5_),(1)
where Y represents the pattern values, (X_1_, X_2_, …, X_5_) represent the measurements of the sensors and the temperature and humidity values, and f represents the convenient functional form of the model.

### 2.3. Methodology

The prediction and model assessment (or validation) are closely related to each other. Particularly, in our task, several models have been considered, of which, those that we have observed to best fit in each case will be analyzed and presented. It is important to mention that, although we have considered different more complex functional forms for the regression models, they have not managed to significantly improve the fits obtained by simple multilinear regression models in all cases. Therefore, the expression of the model used for the fit takes the form:Y = α_0_ + α_1_ X_1_ + α_2_ X_2_ + α_3_ X_3_ + α_4_ X_4_ + α_5_ X_5_,(2)
where α_i_ ∈ ℝ, for i = 0,1, …,5, are the independent term and the contribution of the variables X_i_ in the model. Both fitting to a dataset and choosing the best multilinear regression model can be easily done using the *lm* and *step* functions from the R stats package (there are many works on the internet that show how to do it, such as [49,50]).

In order to evaluate the best fitting model, we have performed the following method. We have split the sample into two disjoint subsets to estimate the prediction error, treating one subset as the training set and the other as the test set (split by vertical lines in Figure 3 and Figure 4). We used the training set to regress each gaseous pollutant on the rest of the variables. Afterward, we predicted a new gaseous pollutant value by applying the fitted model to the new values of the test set. The prediction was compared with the real response value and the prediction ability of the regression model. This provided a measure for the quality of the prediction, which was evaluated by its mean squared prediction error.

#### Training and Test Sets

The methodology applied for each pollutant is similar. Firstly, we evaluate the different regression models using the dataset with all records and choose the one that best fits. Secondly, in order to perform a prediction test, we divide the whole dataset into two subsets: the training dataset and the test dataset. 

The training dataset contained the measurements corresponding to the period from 00:00 h on 08/05/2016 until 08:00 h on 10/05/2016. Thirdly, using this subset, we fit the regression model chosen by fixing the coefficients of the model using the least squares method. The test dataset contained the measurements corresponding to the period from 08:10 h on 10/05/2016 until 09:30 h on 11/05/2016. It is important to mention that the test dataset contained an entire daily cycle, which let us include the possible daily periodicities. Fourthly, with the regression model fitted in the previous phase, we obtain the predictions for the test dataset and compare the results with respect to the pattern values of the test dataset.

## 3. Results

### 3.1. Analysis of Dataset

We can observe in Figure 3a that, in a different proportion, the evolution over time of the measurements taken by the sensor for nitrogen dioxide is closely related and also directly to the pattern values. In addition, in the same sense, we can observe in Figure 3b that there is a high association between ozone measurements, but in this case with an inverse relationship. The previous observations are supported by the correlation coefficients: ρ(O_3_,O_3_s) = −0.8227, ρ(NO_2_,NO_2_s) = 0.6118.

In Figure 3c, we do not observe the existence of an evident relationship between the carbon monoxide measurements captured by the sensor and its corresponding pattern values. In addition, ρ(CO,COs) = −0.3735, which is a low correlation. In line with COs, it is possible to appreciate the existence of a decreasing trend in concentration over time that does not exist in the pattern values curve. In order to better visualize any relationship, we have decided to eliminate the slope of the curve, creating the new variable COsR. However, as we can see in Figure 3d, there is still no evidence of any relationship after removing the slope, and, in this case, an even lower correlation is obtained (ρ(CO,COsR) = −0.1467). We kept the variable COsR in the dataset because the results in the model-fitting work improved.

### 3.2. Fitting Ozone

#### 3.2.1. Selection of the Model

In the case of ozone, first, we considered a multilinear regression model with different combinations among the measurements of the sensors for O_3_s, NO_2_s and COs, in addition to the temperature and humidity measurements. Afterward, we chose the measures of COs instead of its version without a decreasing trend, COsR, obtaining a better adjustment and results. In particular, the model that best fits is:O_3_ = α_0_ + α_1_ COsR + α_2_ NO_2_s + α_3_ O_3_s + α_4_ temp + α_5_ hum,(3)
where α_i_ ∈ ℝ, for i = 0,1, …,5.

Adjusting the model by the least squares method to the dataset with all records, we obtain the α_i_ values contained in Table 2. We observed that all variables considered were significant for the model. In addition, we know that the model manages to explain 75.08% of the total variability of O_3_, and the predictions of the model have a correlation of 0.8665 with the measures of the O_3_ pattern. In the left plot of Figure 5, we compared the values predicted by the model with the measurements of the O_3_ pattern. We can see in the histogram of the net prediction errors of the model that visually these do not differ too much from adjusting to a normal distribution (although the normality hypothesis was rejected when the Shapiro–Wilk test was applied).

#### 3.2.2. Evaluation of the Selected Model

Now, adjusting the model by the least squares method to the training dataset, we obtain the α_i_ values contained in Table 3. We can see that all variables considered in the model are significant and that it manages to explain 71.27% of the total variability of O_3_ for the training dataset. In Figure 6, for the test dataset, we can compare the values predicted by the model with the measurements of the O_3_ pattern and, in the histogram of the net prediction errors of the model, we can observe that these do not differ from a normal distribution. In addition, applying the Shapiro–Wilk test, we obtain a *p*-value of 0.4424, being able to consider the net prediction errors as normal, with mean *μ* = −4.2807 and standard deviation σ = 10.8789. The predictions of the model have a correlation of 0.8824 with the measures of the O_3_ pattern for the test dataset.

### 3.3. Fitting Nitrogen Dioxide

As in the ozone case, firstly we have selected the best fit for nitrogen dioxide, which corresponds to the following multilinear regression model:NO_2_ = α_0_ + α_1_ COs + α_2_ NO_2_s + α_3_ O_3_s + α_4_ temp + α_5_ hum,(4)
where α_i_ ∈ ℝ, for i = 0,1, …,5.

In Table A1 we can see the α_i_ values when we fit the model to the dataset with all records. It can also be seen that all variables are significant for the model, and that the model manages to explain 68.10% of the total variability of NO_2_. The model predictions have a correlation of 0.8252 with the NO_2_ pattern. In Figure A1, it is possible to compare the NO_2_ values predicted by the model with those of the pattern and the histogram of the net prediction errors of the model, which do not differ too much from adjusting to a normal distribution.

To evaluate the chosen model, it was adjusted to the training dataset, obtaining the α_i_ values contained in Table A2. In this case, all variables considered are also significant and it managed to explain 65.55% of the total variability of NO_2_. In Figure A2, for the test dataset, we can see the NO_2_ values predicted by the model, and in the histogram of the net prediction errors of the model, we can see that they also did not differ from a normal distribution. The predictions of the model had a correlation of 0.8301 with the measures of the NO_2_ pattern for the test dataset.

### 3.4. Fitting Carbon Monoxide

The selected model for carbon monoxide has the following expression: CO = α_0_ + α_1_ COsR + α_2_ NO_2_s + α_3_ O_3_s + α_4_ temp + α_5_ hum,(5)
where α_i_ ∈ ℝ, for i = 0,1, …,5.

In this case, adjusting the model to the dataset with all records, the model explains 57.93% of the total variability of CO, and the corresponding predictions have a correlation of 0.7611 with the CO pattern. In Table A3 we can see the α_i_ values that were obtained, all variables being significant. We can also see the predictions of the model and the histogram of its net prediction errors in Figure A3.

Regarding the evaluation of the selected model, once it was adjusted to the training dataset, it explained 47.49% of the total variability and its predictions had a correlation of 0.7769 with the CO pattern. All variables considered are significant for the model, as we can see in Table A4, in addition to the values of the coefficients. In Figure A4, for the test dataset, we can observe the CO values predicted by the model and the histogram of the net prediction errors of the model, which do not differ from a normal distribution.

## 4. Discussion

We observed that all the models generated overcame the global significance contrasts (*p*-values < 0.01) and almost all the individual significance contrasts. In particular, the *p*-values of NO_2_s from Table A3 and Table A4 show that the null hypothesis cannot be rejected by 10% of the significance level (0.09886 and 0.15552, respectively), the reason why the coefficient of NO_2_s in the model of CO is statistically equal to 0. Nevertheless, when this variable is removed from the model, although it simplifies it, neither the adjustment nor the prediction improves. Furthermore, an extension of the dataset will induce the NO_2_ sensor to have a higher influence in the model, providing a better fit for them, as happens with the other pollutants. For this reason, we decided to keep this variable in the model.

Focusing on ozone measurements, and considering all the datasets, the model obtained explains 75.08% of the variability of the data (R-squared), leaving less than 25% to the residuals. We also observed a high direct correlation (0.8665) with the measures of this pollutant pattern. This coefficient indicated a good correspondence between the observations and the predictions of this sensor. Moreover, the histogram of the prediction errors was not normally distributed (Shapiro–Wilk test rejected), although we observed a rough 0 symmetry distribution (Figure 5b).

Nevertheless, when we consider the training dataset for this pollutant, the least square model when adjusted has a lower value of R-squared (71.27%), although it is close to the previous goodness of fit. In this case, the model overcomes the Shapiro–Wilk test on the prediction errors (they follow a normal distribution *n*(−4.28;10.88), *p*-value = 0.4424). It can be seen that 95% of the central prediction errors are between −23 and 12.2 µg/m^3^ with a median of −4.2 µg/m^3^. The interquartile range is 13.1 µg/m^3^. In the boxplot shown in Figure 7a, we observe only 3 outlier values from 337 data points. In Figure 7b, the theoretical normal quantiles, compared to prediction errors, display a good agreement in the central quantiles (points near the straight blue line).

In the case of nitrogen dioxide, we observed that the model obtained explained 68.10% of the variability with a 0.8252 correlation with the NO_2_ pattern. The prediction errors histogram has a slight right asymmetry, although it does not differ excessively from a normal distribution (Figure A1b). Focusing on the training data, we observed a similar value of R-squared (65.55%), and the prediction errors were distributed with the same right asymmetry as before. The mean value of the prediction errors was negative (−11.5 µg/m^3^), with a standard deviation of 12.14 µg/m^3^. These values indicated that the prediction values were greater than the real values, so the model overestimated the NO_2_ values. The asymmetric coefficient is 0.4292, so the distribution shows a right asymmetry with more concentration of negative values of the prediction errors. This bias also shows that the model was overestimating the pattern value measures. We found that 95% of the central prediction errors are between −28.3 and 9.1 µg/m^3^ with a median of −12.6 µg/m^3^. The interquartile range is 16.2 µg/m^3^. Figure 8a shows the rough symmetry of the NO_2_ distribution and Figure 8b presents the deviation of the theoretical normal quantities and the NO_2_ prediction errors.

Regarding the carbon monoxide values, we needed to use the detrended measures of the CO sensor (COs) because the fit is better than COsR. In this case, the variability explained is lower (57.93%) regarding all the data, dropping to 47.49% of the total variability considering the training dataset. Clearly, the distribution of the prediction errors did not follow a normal distribution, with a strong right asymmetry with some values well over 100 µg/m^3^ (asymmetric coefficient: 1.4045). The mean value is 29.84 µg/m^3^ with a standard deviation of 55.51 µg/m^3^. These values indicate that the prediction values were lower than the real ones, so the model underestimated the CO values. Clearly, the adjustment of CO was not as good as the fit of the other pollutants, even after the detrending process. We found that 95% of the central prediction errors were between −34.3 and 164.5 µg/m^3^, with a median of −17.5 µg/m^3^. The interquartile range is also the highest one, with 53 µg/m^3^. Figure 9a shows the clear right asymmetry of the CO distribution and Figure 9b presents the deviation of the theoretical normal quantities and the CO prediction errors.

## 5. Conclusions

In this paper, we present an on-field calibration technique for low-cost MOS sensors, using an adaptive method based on multivariate regression and rigorous statistical analysis. The results show a good adjustment with, at worst, almost 50% of the variability explained by the model. In particular, we found 71.27%, 65.55% and 47.49% of the variability explained for O_3_, NO_2_ and CO, respectively. Considering the short time interval used to estimate the model (less than 2.5 days), and achieving these adjustment values, it is expected that expanding the time series would improve the results.

In the case of O_3_, we obtained the best fit. Ozone prediction errors followed a symmetrical distribution with no bias (the Shapiro–Wilks normality test passed 95% confidence). On the other hand, the NO_2_ and CO prediction errors distribution had a right symmetry, which indicates a greater tail of the distribution in positive values. In these pollutants, the prediction values are generally overestimated with respect to the pattern ones. Overall, we observed a better quality on the fit with higher data.

We observed that the values of CO have the worst fit, which affected the R-squared with the variables considered. To model it, we needed to detrend the sensor measures of monoxide to include them in the calculus. Despite that, the prediction errors were greater than the others, with an average of 29 µg/m^3^ and a marked right bias. We consider that this lack of adjustment in CO was caused by the high time of response of the sensor, the daily variability of this pollutant and the short time interval. Although its calibration may be improved using other more complex models, we consider that for a first approach, the linear multivariate regression is the best-balanced model.

Despite the limitations of the sensors and the dataset used, we obtained a good fit of these gaseous pollutants with respect to the values of the analyzers, while using measurements obtained with low-cost MOS sensors. After the application of our methodology, we observed that the O_3_ and NO_2_ adjusted parameters can be used to give reliable information to citizens and could be used by government agencies for policymaking.

In future works, we will explore other and more complex statistical modeling to enhance the results. We will also verify the possibility of calibrating other MOS sensors through the use of sensors calibrated with the proposed methodology, instead of using control stations. In addition, two of the main disadvantages of the MQ-7 sensor are the delay in the response of the measure and the discontinuous operation mode. In relation to the delay, this is due to the fact that it was designed to measure in ranges 100 times greater than those measured in air quality. Nowadays, new CO sensors working in continuous mode, with the capacity for measuring lower concentrations, has emerged, and these will be considered to replace MQ-7 sensors in future experiments.

## Figures and Tables

**Figure 1 sensors-21-04781-f001:**
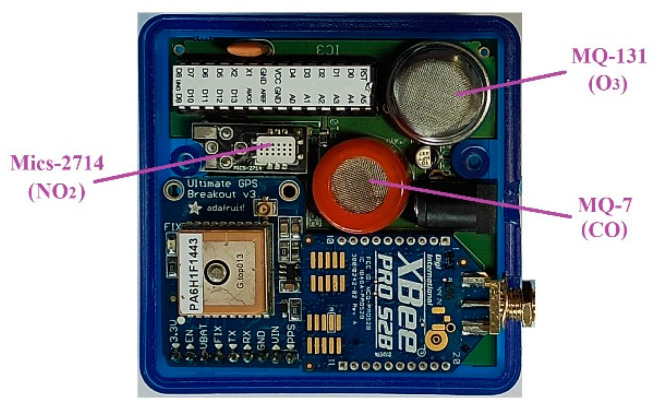
Telemetering devices from the EcoBici project.

**Figure 2 sensors-21-04781-f002:**
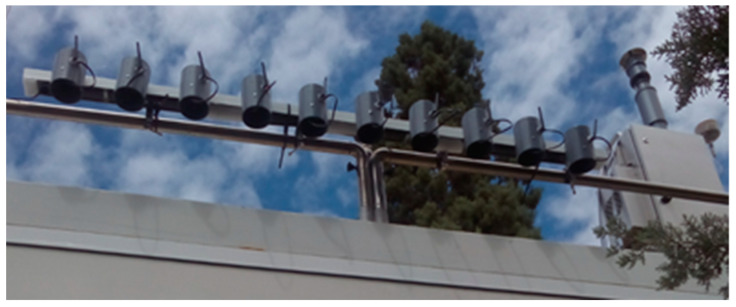
Field calibration of 10 EcoBici devices on the front of an Andalusian Council sampling station in Seville City.

**Figure 3 sensors-21-04781-f003:**
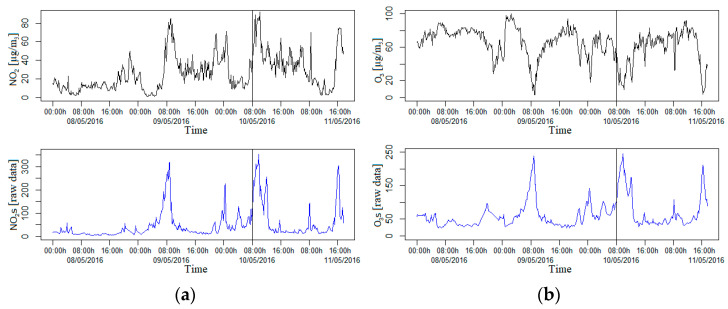
(**a**–**d**) Plots of the evolution over time of the indicated gaseous pollutants in black lines for the patterns and in blue lines for the sensors (and for the transformation of the variable COs, COsR). Each vertical line separates the training dataset (on the left) from the test dataset (on the right).

**Figure 4 sensors-21-04781-f004:**
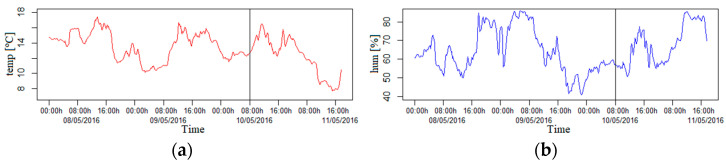
(**a**) The red line corresponds with the evolution over time of the temperature; (**b**) the blue line corresponds with the evolution in time of the humidity. Each vertical line separates the training dataset (on the left) from the test dataset (on the right).

**Figure 5 sensors-21-04781-f005:**
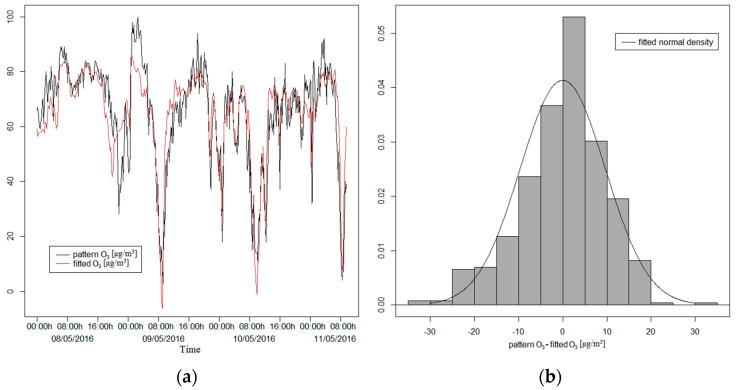
(**a**) The fitted values by the model and the pattern values for ozone; (**b**) the histogram of the net prediction errors of the model.

**Figure 6 sensors-21-04781-f006:**
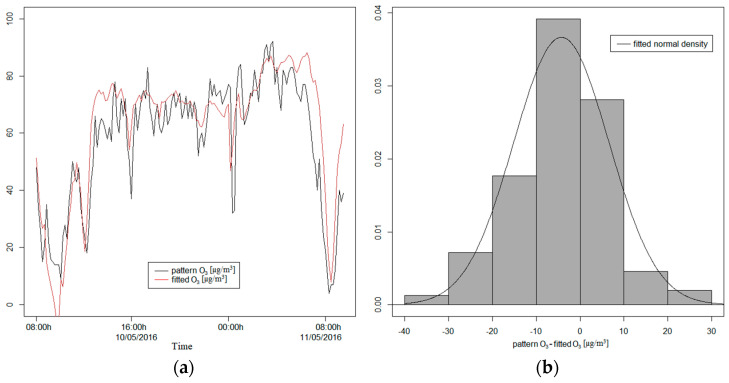
Both graphs with the test dataset. (**a**) The fitted values by the model and the pattern values for ozone; (**b**) the histogram of the net prediction errors of the model.

**Figure 7 sensors-21-04781-f007:**
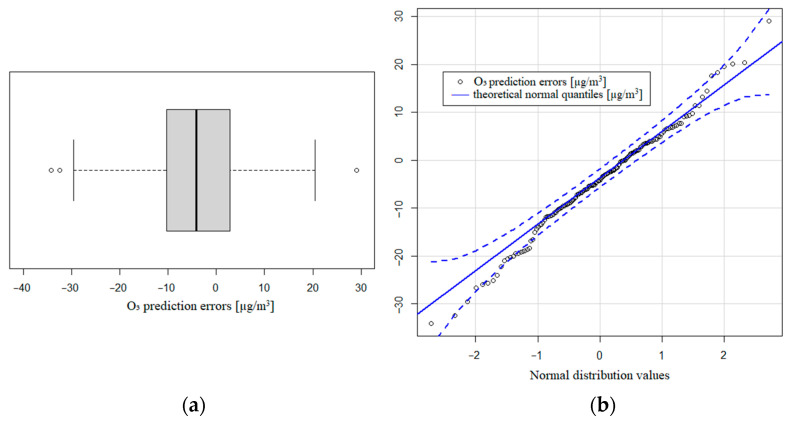
(**a**) Boxplot of O_3_ prediction errors; (**b**) theoretical normal quantiles compared to prediction errors.

**Figure 8 sensors-21-04781-f008:**
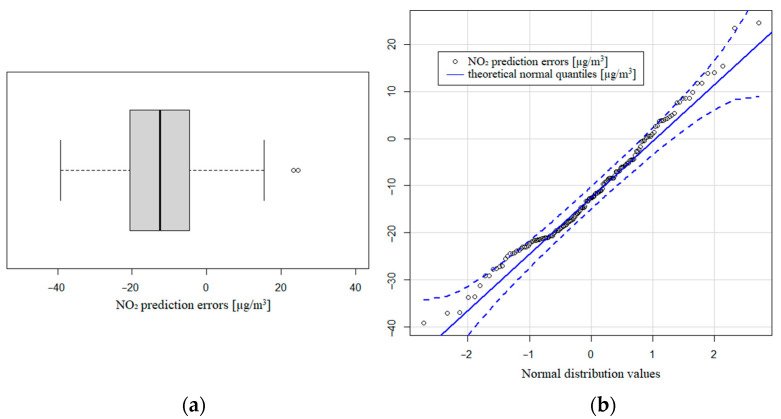
(**a**) Boxplot of NO_2_ prediction errors; (**b**) theoretical normal quantiles compared to prediction errors.

**Figure 9 sensors-21-04781-f009:**
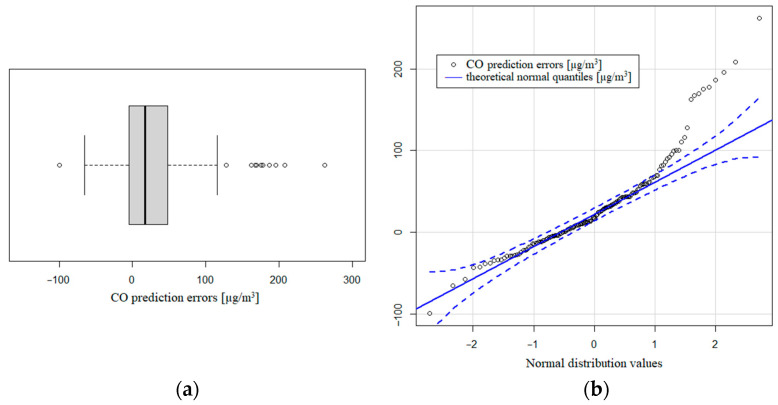
(**a**) Boxplot of CO prediction errors; (**b**) theoretical normal quantiles compared to prediction errors.

**Table 1 sensors-21-04781-t001:** Qualitative levels as referred to for the quantitative levels of each pollutant, and the averaging period used in each of them, following the European EEA standards.

Qualitative Index	SO2 μg/m^3^ (24 h Average Value)	O3 μg/m^3^ (8 h Average Value)	NO2 μg/m^3^ (1 h Average Value)	CO μg/m^3^ (8 h Measured Value)	PM10 μg/m^3^ (24 h Measured Value)
Good	0–63	0–60	0–100	0–5000	0–25
Moderate	63–125	60–120	100–200	5000–10,000	25–50
Poor	125–187	120–180	200–300	10,000–15,000	50–75
Very Poor	>187	>180	>300	>15,000	>75

**Table 2 sensors-21-04781-t002:** The table contains the summary of the model described in Equation (3), adjusted to the dataset with all records. We can see the estimated values and the standard errors of the coefficients, the test statistics and *p*-values of their significance tests (for i = 0,1, …,5, the null hypothesis is α_i_ = 0), the statistics of the residuals and the goodness of fit.

Coefficients	Estimate	Std. Error	*t* Value	*p*-Value
α_0_	−406.43899	54.43049	−7.467	3.85 × 10^−13^
α_1_	0.66569	0.07036	9.461	<2 × 10^−16^
α_2_	0.09424	0.02109	4.468	9.82 × 10^−6^
α_3_	−0.56357	0.03175	−17.752	<2 × 10^−16^
α_4_	−1.01488	0.31333	−3.239	0.00128
α_5_	−0.44478	0.07294	−6.098	2.20 × 10^−9^
Residuals:				
Min	1Q	Median	3Q	Max
−31.110	−5.171	1.232	6.224	30.671
R-squared: 0.7508				

**Table 3 sensors-21-04781-t003:** The table contains the summary of the model described in Equation (3), adjusted to the training dataset.

Coefficients	Estimate	Std. Error	*t* Value	*p*-Value
α_0_	−413.68158	69.18787	−5.979	5.81 × 10^−9^
α_1_	0.69410	0.08865	7.830	6.68 × 10^−14^
α_2_	0.10644	0.02793	3.812	0.000165
α_3_	−0.61144	0.04218	−14.497	<2 × 10^−16^
α_4_	−1.99947	0.42228	−4.735	3.25 × 10^−6^
α_5_	−0.43273	0.08206	−5.274	2.42 × 10^−7^
Residuals:				
Min	1Q	Median	3Q	Max
−32.143	−4.406	1.603	5.626	19.206
R-squared: 0.7127				

## Data Availability

The data presented in this study are available on request from the corresponding author.

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
