# Peer review of "An Approximation for Metal-Oxide Sensor Calibration for Air Quality Monitoring Using Multivariable Statistical Analysis"

_sensors, 2021, doi:10.3390/s21144781_

Round 1

Reviewer 1 Report

The author present an on field calibration technique for low-cost MOS sensors, based on the least squares method to fit regression models, and apply this methodology to fit the pollutant concentrations corresponding to ozone (O₃), nitrogen dioxide (NO₂) and carbon monoxide (CO), though the results are interesting, several issues need to be addressed before publication.

  1. The data acquisition time is from 00:00h of 08/05/2016 to 09:30h of 11/05/2016, only four days, does the author consider multiple collection sites and long-term stability for data acquisition to improved reliability?
  2. Why the author doesn’t consider the influence of other interfering gases in the atmospheric environment on the target gas detection?
  3. In line 485-497 of this paper, the author has found the reason for poor CO fitting. Therefore, the author should increase the time interval, supplement data to make the fitting result of CO more convincing.

Reviewer 2 Report

The main goal of this manuscript was to propose a novel calibration technique for metal oxide sensors (MOS) based on the least squares method using multivariable regression models. This is very important issue on practical application of MOS sensors. Thus, the authors made a very interesting work on it. However, before the manuscript can be considered for publication, some issues on the manuscript must be improved.

The introduction is a bit long and it presents an unbalanced relationship on size and number of citations (long text and few citations). These issues must be improved making it more concise and including more citations.

The last two paragraphs of the introduction are unnecessary. It should be removed or at least well-summarized.

Figure 3 should better elaborate including subitems identification (a, b c…). It is very confused at this way.

The authors have showed single measurements for each sensor. Did they perform reproducibility of the sensor response testing multiples sensors for each gas? And about the stability of the tested sensor response?

Regarding Figure 3 (O3 and O3s), it looks like an opposite sensor response between them (in O3 concentration decreased and in O3s increased at the same time, for instance at 8:00h - 09/05/2016). What is that? Why Does it happen?

Line 262: “Therefore, finally we count 9 variables of work in the dataset”. Which are these variables?

The authors should improve the description of the multivariable regression models including a better description of the used variables and parameters. These information are particularly important to support and allows reproducibility of the proposed model.   

It is suggested to the authors improve the results section. The subsections are very similar (too similar). They may use a supporting information file to include some tables and figures, making the main manuscript text more concise and easy to read.

Reviewer 3 Report

Dear Authors,

The topic of your manuscript is really interesting and important for the impact it could have on our lives. The manuscript deals with the calibration of an array of MOS sensors to predict the value of three pollutants. The paper is simply written, the content and the data treatment are sufficient and sound. However, I suggest that you should make the following suggestions:   

Introduction

o   Lines 39-41: references regarding the reasons why these 3 are the most harmful pollutants could be added;

o   In general, I think that a more concise introduction would increase the readability of the paper. 

Material and Methods

o   Lines 156-165: I think that this part could be moved to Introduction

o   Lines 172-174: this sentence is not completely true, the conductivity changes but after a certain temperature the sensibility to target gases can decrease

o   Line 183: there is no mention regarding the fact that MOS sensors are non-specific sensors

o   Line 184: “de” in “the”

o   Lines 195-199 and 206-214: I think that this part is redundant

o   Lines 217-218: “10 thousand” to “10 000”

o   Lines 228-229: redundant sentence

o   Line 235: why do you talk about Seville if its dataset has been discarded?o   I think that section 2.2.2 should be moved to the results

Results

o   The first paragraph of the “Fitting X” sections describe a general model that should be presented in Material and Methods

o   In general, in the text, figures and table should be described before they are inserted if possible

o   Lines 337 (and others): significance level is chosen?

o   It would be nice to see all sensors’ responses for each target gas since you used all of them for the models and not only the “most specific” sensor

Round 2

Reviewer 3 Report

Dear Authors,

Thank you for following the suggestions. In my opinion, the paper is more readable and clear.

Kind regards.

This manuscript is a resubmission of an earlier submission. The following is a list of the peer review reports and author responses from that submission.

Round 1

Reviewer 1 Report

General Remarks:

The general idea of training the low-cost sensor data by government-regulated and used expensive gas analytical devices from monitoring station data for pollutants such as ozone and NO2 is a good idea, as well as the coupling of this sensor information from different sensor types (Ozone, CO, NO2, Temperature and Humidity) to determine individual pollutant concentrations by multivariable statistical analysis methods. However, the manuscript has clear weaknesses in the presentation of the results, the experimental setup and evaluation.

  • The word aerosol is wrongly used over the whole manuscript, aerosols are small liquids and solids suspended material in air. Neither ozone nor NO2 are liquid or solid in the presented data. The correct terms are gases or pollution.
  • The overall structure of the manuscript should follow the guidelines for authors of the journal (https://www.mdpi.com/journal/sensors/instructions), which means a clearer division of the script into:
    • Introduction
    • Materials and Methods
    • Results
    • Discussion
    • Conclusions
  • Most diagrams do not have any axis titles, like figure 4 till 14
  • For MOS cross sensitivities are usually existent for all mentioned gases. The autors shoul explain whether it plays a role or not
  • Check for consistent use of $XY and XY €
  • Explain training and test data split in figure caption. Line is not explained.
  • line 91 ff.: What ist meant with billions of particles? gases, aerosols, dust? I doubt this statement
  • Most paragraphs contain a lot of unnecessary information, the authors should focus on that what the paper is about
  • Check English, especially singular/plural forms and interpolated clauses
  1. Introduction
  • line 36 NOx and NO2 are doubled (NOx would be sufficient for NO and NO2
  • line 44 ff. paprgraph can be shortend
  • No webpage links should directly placed within the text (line 46, line 47), it should referred to by references
  • More Information about the target gases (Ozone, NO2 and CO) with typical concentration ranges should be mentioned.
  • When combining sensory data of NO2 and Ozone for determination of the concentrations of these gases, the Leighton relationship should be mentioned since it describes the reactions between these gases in our atmosphere during daytime.
  1. Sensors
  • The units µg/m³ and ppb, are common knowledge and do not need to be introduced and discussed in detail are in the lines 135-153, in general the paprgraph shoul be shortened up to line 168
  • line 188ff. mention the manufacturer in the text
  • Figure 2 has a very bad resolution and look like graphs from the data sheets of the sensors and not measured data. Therefor it should be clearly referred to by references and do not need to be shown as graphs inside the manuscript.
  • A large part of the information shown is background information about the gases and sensor types that should be integrated into the introduction section.
  • It becomes not clear, how big the distances between the location of reference data and the location of gas sensor data is. Since each investigated pollution is highly variable distribute within one city and can enormously differ between two city caused by wind, weather, sunlight, geological condition, industry, degree of buildings and green areas, it is necessary to have a close distance between the sensors and the reference analytical systems. If the presented reference data is originated in Seville City and the sensor data is coming from Granada, a distance of nearly 250 km makes the data not comparable.
  • line 219 till end of chapter can be shortened, it contains a lot of unnecessary information regarding the topic of the paper
  1. Data
  • Line 263: the gases are no aerosols!
  • Figure 5: y-axis descriptions are missing, see also all other figures
  • Which software was used for data analysis?
  • using COsR: how did you eleminate the slope to compensate the drift? Did you check wether curve smothing/averaging (maybe in combination with the first deviation) will yield better results?
  1. Methodology
  • Subchapter 4.1 Nearly the complete chapter is common knowledge, not necessary for the content of this paper or should be part of the introduction. Furtheron it becomes not clear how the data were calculated using the given syntax and the introduced functions seem not to be used anymore in the following chapters
  • Subchapter 4.2. The advantage of a test and training approach is that it allows the robustness of a model to be checked using an independent set of test data. Thus, finding/building a model should be done directly with the training data set (in this case data from 00:00h on 08/05/2016 until 08:00h on 10/05/2016) and the best fit solution should then be checked with the independent test data set ( from 08:10h on 10/05/2016 until 09:30h on 11/05/2016). Whether this has been done in this way is not entirely clear and should be explained more precisely.
  1. Adjustment results
  • Table 1 to 6 are repetitive and could be merged to a smaller number of tables as well as Figures 6 to 11 and equations 5.10 to 5.18. The presentation is repetitive and not precise therefor hard to read and to this extent not necessary. (e.g. equation 5.11 and Table 1 show the same). In the table the corresponding αi should (also) be mentioned
  • t value and p value were not defined
  1. Analysis of results
  • Line 475: What is the Null Hypothesis?
  1. Conclusion
  • The conclusion is more a summary of results. The authors should give a suggestion how this methods can be adapted to sensor systems, which are not nearby the control stations. It remains open whether the models can be transferred directly to other sensor nodes (having different charges of MOS sensors) located in other environmental settings (especially for longtime measurements). Can a model, created by one sensor node located near the control station, directly be transferred to other sensor nodes at different locations (see also 4. second bullet point)?

Reviewer 2 Report

As a resume, the main results of the work are the following three equations,

O₃=       -413.68158 +0.6941 COsR   +0.10644 NOâ‚‚s        -0.61144 O₃s   -1.99947 temp    -0.43273 hum.        (5.12)

NOâ‚‚=     306.28198  -0.56296 COs         -12499 NOâ‚‚s      +0.53380 O₃s  +6.29591 temp  +0.72325 hum.        (5.15)

CO=        4297.8297     -5.2999 COs      +0.1904 NOâ‚‚s      +0.85370 O₃s     +5.7620 temp     +2.0691 hum.        (5.18)

Analyzing the three equations the dependence with the temperature and humidity are completely different of what can be expected regarding the data-sheets of the sensors.

The authors doesn't give any explanation regarding the strong dependence of the each gas concentration with the other two gas analyzed, a such so strong dependence are not reported in the data sheet.

In the selection of the data it is assumed that all the gas concentrations are expressed in µgr/m3 and there are two things unacceptable,

first: O3s is supposed to be the concentration values of ozone recollected by sensor MQ-131 and expressed in µgr/m3, so it is unacceptable to invert de data, concentration with negative values has no sense.

second: COs is supposed to be the values of the sensor MQ-7 expressed in µgr/m3 , it seems that data shows a drift with the time, much larger than expected reading the corresponding data sheet.

All that seems indicated that the sensors doesn’t work well. It has no sense that the real value of the ozone concentration depends more on the measurements of the CO sensor than the ozone sensor, or that the rapport between the real value of the NO2 concentration and the value of NO2 sensor is 12499.

Reviewer 3 Report

The paper, "An approximation for metal-oxide sensor calibration for air
quality monitoring using multivariable statistical analysis" presents a methodology for low-cost sensors calibration. My comments are given below. The paper is not ready to be published in a peer review journal and requires some major improvements. 

Abstract

Line 13: define MOS, abstract should be able to stand alone

Line 14 – 16: “usually employed to measure gases in high concentrations, and their use  for low values is only valid through exhaustive calibration process and subsequent precision analysis”. Does this mean, no calibration is required when concentrations is high? This is not correct. Low cost sensors require calibration and precision analysis regardless whether the concentration is high or low.

Line 16: “new on field calibration technique”, what is new about this calibration, the OLS model?

Some results should be presented in the abstract. You abstract is just made of introduction and methodology, there are no results and conclusion. It must be improved.

The key words “multivariable statistical analysis; multi- variable regression models” are almost the same thing. I suggest to use one of them and add “calibration” or “model calibration”.

Introduction

Line 27 – 29: “Air quality is essential for human being and the environment. It affects our health in a direct way, since it is what we breath, but also in an indirect way, since it has a great impact in vegetation, fauna, and the quality of water and soil, i.e. what we drink and eat”. There are several problems with this sentence. I will point out some of the issue with this sentence and request the authors to revise the whole paper for such shortcomings. “air quality is essential” is not correct, it should be either “clean air” is essential or “Good air quality is essential”. “It affects our health …. It is what we breath”, we don’t breath air quality, we breath “air”. “Vegetation, fauna”, it should flora and fauna…. If you are talking about flora and fauna then you should carry on water and soil. “i.e. what we drink and eat”, we don’t drink and eat soil. This is an example of bad structure, you should revise the whole paper for such weakness.

Line 33: 400.000 premature deaths, is it 400 or 400 thousand? If 400 please remove the extra zeros.

Line 35: “parameters”, these are pollutants. Also I don’t think CO2 is considered a pollutant, it is rather considered a greenhouse gas. These are minor points but we need to be very clear.

Line 40: “specifically PM10, which are directly related to traffic”, better say PM2.5.  Recent studies show that PM10 is not directly related with road traffic.

Line 46 and 47: provide a number in bracket and move the website to the reference list.

Line 52: “carry out accurate models” …. Modelling

Line 54: “emissions”, I think it should be concentrations.

Line 57: “face”…. Should be “address”.

Line 63: “as previously mentioned above”, delete this.

Line 78: 100€ (~100$), I think this price for EC sensors is too low. AQMesh, Zephyr and Envirowatch are EC sensors are their price is in £3 to 6 k. $100 is noting. If you have reference please provide a reference a name of the EC sensors here.

Line 125: “the model fits and the predictions”.. to fit and validate the model.

Line 133 – 134: “we apply it to fit the aerosols: ozone, nitrogen dioxide and carbon monoxide”, please revise this sentence, it does make senses. Also write the short names of these pollutants. You have used colon (:) after aerosols, which reads like the aerosols are O3, NO2 ……

line 136: “pollutants” should be pollutant concentrations

Line 136 – 142: ppb or ug/m3  is a commonly known thing, I suggest to remove this part.

Line 144 – 153: This is not correct and is not required, please remove it.

There is a clear difference between concentrations and emissions. You need to clarify these, I don’t think the authors really understand the difference.

Line 169 – 179: provide reference

Line 187 – 188: these pollutants have been mentioned above and should be defined at first appearance in the text. Here, only the formulas should be used.

I don’t know what the authors mean by aerosol. Aerosol are basically tiny particles in the atmosphere.

I did not read the paper in details beyond line 188, just had a quick look through. The papers has several limitations. I mentioned a few them below.

Major

There are so many details which are not required here.

Language needs significant improvement

To assess the model performance generally we use correlation coefficients, RMSE, R2 and several other metrics. Herehey have used only r-squared value.

The data is very limited just for a few days. Which is further divided into training and testing dataset. The testing and training datasets should randomly selected.

They have employed very simple OLS regression model, nothing is new about the methodology.